# Brewer’s Spent Yeast Cell Wall Polysaccharides as Vegan and Clean Label Additives for Mayonnaise Formulation

**DOI:** 10.3390/molecules28083540

**Published:** 2023-04-17

**Authors:** Sofia F. Reis, Pedro A. R. Fernandes, Vítor J. Martins, Sara Gonçalves, Luís P. Ferreira, Vítor M. Gaspar, Diogo Figueira, Diogo Castelo-Branco, João F. Mano, Manuel A. Coimbra, Elisabete Coelho

**Affiliations:** 1LAQV-REQUIMTE, Department of Chemistry, University of Aveiro, 3810-193 Aveiro, Portugal; sreis@ua.pt (S.F.R.); pedroantonio@ua.pt (P.A.R.F.); vitorjmartins@ua.pt (V.J.M.); sarafgoncalves@ua.pt (S.G.); mac@ua.pt (M.A.C.); 2Department of Chemistry, CICECO—Aveiro Institute of Materials, University of Aveiro, 3810-193 Aveiro, Portugal; lpferreira@ua.pt (L.P.F.); vm.gaspar@ua.pt (V.M.G.); jmano@ua.pt (J.F.M.); 3Mendes Gonçalves SA, Zona Industrial, Lote 6, 2154-909 Golegã, Portugal; diogo.figueira@casamg.pt (D.F.); diogo.castelo.branco@mendesgoncalves.pt (D.C.-B.)

**Keywords:** mannoproteins, glucans, emulsions, sauces, glycosidic linkages, droplets distribution, back-extrusion

## Abstract

Brewer’s spent yeast (BSY) mannoproteins have been reported to possess thickening and emulsifying properties. The commercial interest in yeast mannoproteins might be boosted considering the consolidation of their properties supported by structure/function relationships. This work aimed to attest the use of extracted BSY mannoproteins as a clean label and vegan source of ingredients for the replacement of food additives and protein from animal sources. To achieve this, structure/function relationships were performed by isolating polysaccharides with distinct structural features from BSY, either by using alkaline extraction (mild treatment) or subcritical water extraction (SWE) using microwave technology (hard treatment), and assessment of their emulsifying properties. Alkaline extractions solubilized mostly highly branched mannoproteins (*N*-linked type; 75%) and glycogen (25%), while SWE solubilized mannoproteins with short mannan chains (*O*-linked type; 55%) and (1→4)- and (β1→3)-linked glucans, 33 and 12%, respectively. Extracts with high protein content yielded the most stable emulsions obtained by hand shaking, while the extracts composed of short chain mannans and β-glucans yielded the best emulsions by using ultraturrax stirring. β-Glucans and *O*-linked mannoproteins were found to contribute to emulsion stability by preventing Ostwald ripening. When applied in mayonnaise model emulsions, BSY extracts presented higher stability and yet similar texture properties as the reference emulsifiers. When used in a mayonnaise formulation, the BSY extracts were also able to replace egg yolk and modified starch (E1422) at 1/3 of their concentration. This shows that BSY alkali soluble mannoproteins and subcritical water extracted β-glucans can be used as replacers of animal protein and additives in sauces.

## 1. Introduction

Emulsions are colloidal dispersions composed of at least two immiscible fluids, one dispersed into the other, one in the form of small droplets. Water is the continuous phase in oil-in-water emulsions such as mayonnaise, where the fat globules are suspended in an aqueous phase that contains natural emulsifiers, as proteins and carbohydrates [1]. Emulsifiers are compounds capable of improving the stability of a mixture of two immiscible liquids by promoting the solubility of substances through the formation of emulsions. In this sense, emulsifiers are largely used in several industrial fields. In the case of food industry, emulsifiers take part in the formulation of sauces, contributing to their texture, flavor, and aroma. Among these, lecithin, an emulsifier found in egg yolks that allows the dispersion of oil-in-water, is often used in sauces like mayonnaise [2]. Proteins, as those derived from egg white and milk, are also used as emulsifiers, where the balance between hydrophilic and hydrophobic amino acids and protein structural conformation govern their emulsification performance [3]. However, these emulsifiers are usually prone to yield unstable emulsions, especially near their isoelectric point or in the presence of salts. For this reason, emulsions often present stabilizing agents such as xanthan gum [4] and starch [5] to inhibit the coalescence of emulsions. Most of these stabilizers are labeled as food additives, and thus are not positively perceived by consumers. Besides, some consumers are looking for the replacement of animal protein by other sources of protein [6]. These demands are supported by a rationale of improved naturalness and sustainability of food products, established by the clean label concept. In this context, industries are on the hunt for new ingredients capable of fulfilling consumer expectations [7].

Yeast components have been identified as efficient compounds with emulsifying properties [8,9,10,11]. These included mannoproteins, which are polysaccharides located at the surface of the yeast cell wall, accounting for 30–50% of its dry weight [12]. Mannoproteins differ in their mannan to protein ratio, branching degree, phosphorylation, or substitution via glycosylphosphatidylinositol anchors to β-glucan chains [13]. Besides intra strains/species, mannoproteins’ structural features are also affected by the extraction process, which includes autoclave [9], enzyme [9], or alkali extractions [13,14]. Due to this diversity, mannoproteins are ascribed to a wide variable emulsifying performances that, together with the absence of established extraction processes with good yields, have impaired their commercial exploitation as emulsifiers [15]. The commercial interest in yeast mannoproteins might be boosted considering the consolidation of their properties in a range of environmental conditions supported by structure/function relationships. In this sense, this work aimed to attest the emulsifying behavior of yeast mannoproteins with distinct structural features. To achieve this, brewer’s spent yeast (BSY), the second major by-product of the brewing industry [16], was used as a low-cost and highly available source of mannoproteins [14,17] under a circular economy concept. To obtain polysaccharides of distinct structural features, subcritical water extraction using a microwave was performed as a harsher cell wall extraction process [18,19], while alkali extraction was performed as a milder treatment [14], to obtain more preserved polysaccharide structures. The resultant extracts were then tested regarding their emulsifying behavior in a water and vegetable oil emulsion, in a mayonnaise model emulsion, and as part of mayonnaise formulations.

## 2. Results

### 2.1. BSY Extracts Composition

The autolysis precipitate of brewer’s spent yeast (BSY) was composed of 30% polysaccharides rich in Glc (70 mol%), outcoming from glucans, and Man (30 mol%), from mannoproteins (Table 1). The yeast cell walls were then purified using an aqueous 80% ethanol solution, allowing for the obtainment of an alcohol insoluble residue (AIR) enriched in polymeric material. The alkaline extraction with 1 M KOH, after neutralization and dialysis (12 kDa cut-off) steps, allowed to obtain three fractions: (1) 12% of a soluble fraction (Sn 1M) composed of 69% carbohydrates (Table 1), mostly highly branched mannoproteins (66 mol% Man) (Table 2), as shown by the prevalence of 2-Man (16 mol%) and 2,6-Man (11 mol%), with a protein content of 23%; (2) 3% of a precipitate (pp 1M) composed of 16% carbohydrates, mainly glycogen and (β1→3)-glucans, as shown by the content of 4-Glc (48 mol%), 4,6-Glc (2 mol%) and 3-Glc (18 mol%), and 27% protein; (3) 3% of a soluble material obtained from the final residue during the dialysis process (Sn_FR 1M), rich in carbohydrates (60%) and 14% of protein, probably accounting for glycogen, given the 4-Glc (45 mol%) and 4,6-Glc (2 mol%) diagnostic linkages.

The extraction with 4 M KOH, in line with the previously described 1 M KOH extraction, allowed to obtain the same three extracts (Table 1) with similar yields, except for the precipitate (pp 4M), which attained a higher recovery (13%) from the autolysis precipitate than pp 1M (3%) and presented a higher protein proportion (73%). The fractions Sn 4M and pp 4M have similar chemical composition and glycosidic linkages to those obtained with 1M KOH (Table 2). However, the soluble material obtained from the dialysis of final residue (Sn_RF 4M) presented 84 mol% of glucans, as opposed to Sn_RF_1M, which presented 54 mol%.

Subcritical water extractions (SWE) using a microwave at 180 and 200 °C allowed for the solubilization of 32 and 42% of BSY cell wall material (Table 1). Around 23% out of the solubilized material accounted for the polymeric material that precipitated in solutions of 80% ethanol (pp 180 °C). This material was found to be mostly composed of carbohydrates with an equimolar proportion between Man and Glc (Table 1) that suggested a mannoproteins and glucans proportion of ≈1:1. Glycosidic linkage analysis (Table 2) showed a high prevalence of 2-Man (10–13 mol%) and 3-Man (3–4 mol%), as well as terminally linked (t-Man, 24–25 mol%), suggesting mannoproteins of short mannan chains. The extracted glucans presented 4-Glc (24–34 mol%), 3-Glc (3–18 mol%) linkages, and residual 4,6-Glc and 3,6-Glc linkages, suggesting the presence of low branched or linear glucans.

The SWE material that remained soluble in ethanol (Sn) was found similar protein content and to be composed of 15% and 38% of carbohydrates in SWE 180 °C and 200 °C treatments, respectively. The occurrence of free sugars (5%), resultant from depolymerization reactions during microwave extraction, and the solubility in ethanol suggests that these glucans and mannoproteins were of low molecular weight. The difference between the two treatments was the higher content of 3-Glc linkages extracted when applying SWE at 200 °C than at 180 °C.

### 2.2. Emulsifying Properties of BSY Extracts

The emulsifying capacity and stability of oil-in-water emulsions of BSY extracts were assessed when prepared by mixing water and vegetable oil (4:6) using hand-shaken or ultraturrax stirring methods. The structure–function relationship on selected emulsions was achieved regarding the polysaccharide structures involved on the emulsion formation and droplet size distribution.

#### 2.2.1. Emulsifying Capacity and Stability

The BSY extracts revealed different emulsifying capacities (Figure 1) and different performances when hand-shaken or stirred using ultraturrax, suggesting an emulsion dependence on the energy applied. Some of the BSY extracts have no emulsifying capacity, such was the cases of pp 1M, Sn 180 °C, and Sn 200 °C. In the case of pp 1M and Sn 180 °C, only half of the extracted material was composed of carbohydrates and protein. In the case of Sn 180 °C and Sn 200 °C, the glucans and mannoproteins present were of low molecular weight, which may explain the absence of emulsifying properties. When hand-shaken, the extract which revealed the highest emulsifying capacity was pp 4M (63%), yet when using ultraturrax, it was pp 200 °C (72%). Xanthan gum presented the highest emulsifying capacity (94%) only when using ultraturrax, possibly because the energy applied by hand-shaking was not enough to form an emulsion.

The emulsifying capacity of Sn 1M was three-fold higher when using ultraturrax than hand-shaking, yielding emulsions that were stable for at least 1 month (Figure 1). The material solubilized from BSY using higher alkalinity (Sn 4M) presented similar emulsifying capacity by hand-shaking or ultraturrax, with the hand-shaken emulsion being the more stable one. The emulsifying capacity of the Sn_FR 1M was two-fold higher when using ultraturrax than when hand-shaken, losing only 10% of the emulsion volume after 1 month. The emulsion capacity of Sn_FR 4M was two-fold higher and more stable when hand-shaken. The pp 4M was only 1.4-fold higher when hand-shaken compared to ultraturrax, being more stable when hand-shaken. The emulsifying capacity of pp 180 °C was three-fold higher when hand-shaken. However, it lost half of the emulsion after 1 month, with its emulsifying capacity being similar when using ultraturrax. In contrast, the emulsifying capacity of pp 200 °C was four-fold higher when using ultraturrax and lost only 5% emulsion after 1 month, a loss similar to xanthan gum (8%).

The highest emulsifying capacity of pp 4M when hand-shaken seems to be related with the protein content (Table 1), while the highest emulsifying capacity of pp 200 °C when using ultraturrax seems to be related with the carbohydrate content. The different emulsifying performances of Sn 1M and Sn 4M could be due to their different glycosidic linkage patterns (Table 2), with the Sn 1M presenting a higher proportion of 2,6-Man (11 mol%) than Sn 4M (4 mol%). Additionally, Sn 4M was more balanced in mannoproteins and glucans content than Sn 1M. In the case of Sn_FR 4M, its similar glycosidic composition (Table 2) to pp 4M may explain its best performance when hand-shaken.

#### 2.2.2. Emulsifying Polysaccharides of BSY Extract’s Emulsions

To understand the differences observed in the emulsifying capacity between extracts, the polysaccharides present in the emulsions and in the separated aqueous fractions obtained after a 1-month storage period were analyzed. The emulsions chosen were the ones that performed the best when hand-shaken (pp 4M) and when using ultraturrax (pp 200 °C). Additionally, the ones with similar composition but with different performances, Sn 1M and Sn 4M, were chosen over to Sn_FR 1M and Sn_FR 4M, because of their higher representativeness in the BSY solubilized extracts (Table 1).

The aqueous phases separated after emulsion destabilization from the hand-shaken emulsions were higher in pp 200 °C, Sn1M, and Sn 4 M than in pp 4M, while from the ultraturrax emulsions were higher in Sn 1M and Sn 4 M than in pp 4M and pp 200 °C (Table 3), reflecting their emulsifying capacity and/or stability (Figure 1). After a liquid-liquid extraction (LLE), allowing the partition of the compounds that are contained in the emulsion phases, all the hand-shaken emulsion material was extracted into the water phase, except for pp 4M emulsion, where 25% remained at the hexane phase, while the ultraturrax emulsion material was divided by the two phases (Table 3). The ultraturrax emulsions, according to the lower amount of material, which remained in the aqueous phase upon the phase separation after 1-month storage, and to the higher emulsion material soluble in the hexane used in the LLE, seem to be more stable than hand-shaken emulsions.

The polysaccharides present in the pp 4M emulsion phase obtained by hand-shaking, which remained after 1 month (Table 4), comprehended linear or low branched 4-linked glucans, belonging to glycogen (65 mol% 4-Glc; 4 mol% 4,6-Glc). (β1→3)-Glucans (5 mol%), absent in the polysaccharides present in the aqueous phase, were also detected. Ultraturrax (Table 5) promoted the presence of short chain mannans seen by the increasing amount of t-Man (5 to 18 mol%) and the presence of (β1→3)-glucans, linkages absent in the aqueous phase. These results suggest the ability of (β1→3)-glucans to be dispersed in the continuous phase, which may increase the density of the aqueous phase to values near to the density of the droplets, an important characteristic to the emulsion stability.

In the case of the pp 200 °C emulsion, the composition in glucans (59 mol%), accounting low branched (1→4)- and (β1→3)-glucans in a proportion of 2:1, and mannoproteins (41 mol%), was balanced when using hand-shaking (Table 4). However, when applying ultraturrax (Table 5), the proportion of mannoproteins increased, being also composed of short chain (α1→2)-mannans. This process also increased the proportion of (β1→3)-glucans to (1→4)-glucans.

Sn 4M emulsion by hand-shaking (Table 4) was also balanced in glucans (49%) and mannoproteins (51%), while ultraturrax (Table 5) promoted the increase of short chain mannans to 81% in emulsion. The polysaccharides present in Sn 4M emulsion were mainly (α1→2)-mannans and a small amount of (1→4)-glucans derived from glycogen. Mannans linkage composition was similar when compared to pp 200 °C emulsion, but the glucans present in emulsion were different. In emulsions produced with pp 200 °C the glucans structure that are more prevalent in emulsion were the (β1→3)-glucans, which may explain the difference between these two extracts in the stability when using ultraturrax. The absence of (β1→3)-glucans in Sn 4M emulsions may explain the low stability when using ultraturrax (Figure 1). In the case of the Sn 1M emulsion, a high proportion of mannoproteins (78 mol%) was observed, while (β1→3)-glucans (6 mol%) were present in low amounts, explaining the higher emulsion capacity and stability than Sn 4M emulsion when using ultraturrax.

#### 2.2.3. Emulsions Droplet Size Distribution

The BSY extracts that exhibit the best emulsifier properties (Section 2.2.1) were selected for microscopy analysis and droplet size distribution assessment. The extract pp 4M (best emulsifier by hand-shaking—Figure 1a), the extract pp 200 °C (best emulsifier by ultraturrax—Figure 1b), and the xanthan gum, the reference emulsifier, were analyzed after 1 month by widefield fluorescence microscopy (Figure 2). The three samples revealed polydisperse emulsions [1].

The normal distribution of droplets size calculated from the respective optical contrast micrographs (Figure 2) showed that pp 4M emulsion presented the smallest size droplets population, with a range of 4–41 μm, while pp 200 °C and xanthan gum presented a similar population of droplets size with ranges of 5–192 μm and 7–178 μm, respectively. This may explain the higher stability of pp 4M emulsion when comparing to the pp 200 °C and xanthan gum emulsions (Figure 1), since emulsion stability is related to the small droplets size. Thus, despite pp 200 °C and xanthan gum emulsions presented higher emulsifying capacity than pp 4M, they are likely to be more susceptible to coalescence (Oswald ripening). At 1-month storage, the highest frequency of droplets size, in both pp 200 °C and xanthan gum emulsions, was between 10–30 μm (Figure 2b,c), however they presented a wide range of droplets size distribution being the widest range in xanthan gum emulsions (Figure 2c) despite presenting the highest emulsifying capacity.

### 2.3. Emulsifying Properties of Mayonnaise Emulsion Models with Addition of BSY Extracts

The emulsion capacity and stability of the BSY extracts were measured in a mayonnaise model solution (pH 3) and compared to common emulsifying food additives as xanthan gum, a modified starch, and freeze-dried egg yolk (Figure 3), usually used in mayonnaise formulations. Emulsions were prepared by ultraturrax stirring (13,500 rpm) also in line with standard mayonnaises preparation.

At 24 h, xanthan gum presented the highest emulsifying capacity (100%) followed by pp 4M (84%), freeze-dried egg yolk (74%), pp 200 °C (73%), modified starch (64%), Sn 4M (47%), and Sn 1M (29%). However, after 38 days, the highest emulsifying capacity was presented by pp 4M (80%) and pp 200 °C (71%), while xanthan gum lost 36% of its emulsifying capacity, modified starch lost 10%, and freeze-dried egg yolk lost 7%. The BSY extracts Sn 1M and Sn 4M presented the lowest emulsifying capacity after 38 days, with 12% and 8%, respectively (Figure 3). Comparing the emulsifying capacity of these experiments with those at neutral pH (Figure 1b), it seems that, by decreasing pH, the emulsion performance of pp 4M (44% to 82%) and Sn 4M (28% to 52%) increased by almost the double. In contrast, for Sn 4M, the stability decreased, since at 38 days lost 44% of its emulsifying capacity. In the case of pp 200 °C and Sn 1M extracts, pH had no influence on the emulsifying capacity, but in the case of Sn 1M it affected its stability, decreasing from 33% to 12% at acidic pH. Xanthan gum decreased its emulsifying capacity from 94% to 84% when decreasing pH from neutral to acidic, also decreasing emulsion stability. At neutral pH, xanthan gum lost 8% of its emulsifying capacity after 30 days while at acidic pH lost 25% after 38 days.

The texture of the emulsions immediately after stirring was evaluated by determining firmness, consistency, cohesiveness, and work of cohesion, following a back-extrusion test using a texturometer. Xanthan gum emulsions revealed the highest firmness (35 g), consistency (875 g·s), cohesiveness (−34 g) and work of cohesion (−48 g·s), at least 4-fold higher than those observed for freeze-dried egg yolk and modified starch (Appendix A). At the same concentration (2.5 mg/mL), all BSY extracts showed close values of firmness, consistency, cohesiveness, and work of cohesion values, to that observed for freeze-dried egg yolk and modified starch (Figure 4; Appendix A).

### 2.4. Mayonnaises Formulation Using BSY Extracts

BSY extracts were used as vegan and clean label substitutes for egg yolks and modified starch in mayonnaise formulations. The pp 4M, which revealed the best emulsifying capacity and stability in the mayonnaise emulsion models, was firstly applied as a replacer of egg yolk and modified starch in mayonnaise formulations, on a dry weight basis, added up to a concentration of 3%. This resulted in a very thick texture, characteristic of a pudding, not allowing to be accurately measured with the protocol established for measuring mayonnaise texture. Thus, the pp 4M addition was decreased to a concentration of 1%, which allowed to obtain a more fluid mayonnaise with firmness (63 g), consistency (1723 g·s), cohesiveness (−63 g), and work of cohesion (−87 g·s), close to the values of the standard formulation (Figure 5; Appendix A). Sn 1M, Sn 4M, and pp 200 °C also resulted in mayonnaises when used at the same concentration (1%)**.** However, the firmness (11 to 14 g), consistency (171 to 220 g·s), cohesiveness (−10 to −16 g) and work of cohesion (−2 to −8 g·s) conferred by these extracts was almost one order of magnitude lower than that reported for the standard formulation.

## 3. Discussion

Although BSY is a by-product rich in mannoproteins and glucans, in *S. pastorianus* only mannoproteins can be extracted by conventional methods [14,17]. Alkaline 1 M and 4 M KOH extractions solubilized mostly highly branched mannoproteins, a structural feature typical of the *N*-linked type [13], alongside with glycogen. A protein extract insoluble in water at neutral pH, composed mostly of *O*-linked mannoproteins, was also obtained when extracting with 4 M KOH (pp 4M). Subcritical water extraction (SWE) using microwaves allowed to extract BSY components at comparable yields to those reported (32–42%) for the extraction from wine lees using autoclave treatment [9]. The subcritical water provides an autohydrolysis of cell wall components allowing to co-extract glucans together with mannoproteins. The polymeric material (pp 180 °C and pp 200 °C) isolated from the solubilized components was composed of mannoproteins with short mannan chains with a glycosidic linkage pattern characteristic of *O*-linked mannoproteins [13], alongside with (1→4)- and (β1→3)-linked glucans. Overall, it could be concluded that SWE yielded extracts with a mix composition in polysaccharides while the alkali extractions represent a more selective approach.

The emulsifying capacity achieved by the BSY extracts at neutral pH was different and dependent on the energy applied. Hand-shaking was enough to achieve a good emulsion when the extracts have a high protein content (pp 4M). However, when using ultraturrax stirring, the best emulsion capacity was achieved with pp 200 °C, with higher proportion of short chain mannans and β-glucans, found to be very relevant for the emulsion stability. The best emulsifying capacity performance of pp 4M when hand-shaking seems to be driven by the amount of protein and the presence of glycogen and/or linear (1→4)- and (β1→3)-glucans. (β1→3)-Glucans are responsible for the emulsion stability, when using both hand-shaking or ultraturrax. These ones, simultaneously with the short chain mannans, are responsible for the performance of emulsifying capacity achieved when using ultraturrax. The best emulsifying capacity performance of pp 200 °C when using ultraturrax seems to be driven by the content of (β1→3)-glucans, since ultraturrax promotes the increase of mannoproteins composed of short chain mannans, as well as the increase of (β1→3)-glucans in the emulsion. Both polysaccharides are important for emulsion formation and the latter for the emulsion stability as well. β-Glucans are known to present a very high water retention capacity (≈4 g of water/g) [20], turning emulsions more viscous and stable. However, emulsion stability was found not to be related to emulsion capacity, due to the different susceptibility to Ostwald ripening of each extract, where small droplets in the emulsion system increase the size of large droplets at the expense of themselves. This is due to the higher solubility of oil molecules in the vicinity of the small droplets when compared to the larger ones [5]. The pp 200 °C and the reference emulsifier xanthan gum, despite having better emulsion capacity than pp 4M when using ultraturrax stirring, were less stable. Both extracts produced polydisperse emulsions [1] and the normal distribution of droplets size of BSY emulsions showed values within the ones reported for other yeast mannoproteins: 4.3–156 μm from *Saccharomyces cerevisiae* [10] and 40–242 μm from yeast isolated from sugar palm wine [21]. However, emulsions produced using pp 4M presented the smallest droplets size after 1-month storage. The highest stability of this extract is explained by the presence of small droplets [1,5] and also to the high protein content. This enhances the amphiphilic character of mannoproteins and, hence, their ability to decrease the surface tension at the interface of the two immiscible phases [10].

The BSY extracts pp 4M and pp 200 °C also presented higher stability at acidic pH (pH 3) and ultraturrax stirring when compared to the reference emulsifiers, possibly due to the lower solubility of xanthan gum (p*Ka* of 3.1) and poor stability of lecithin, the main emulsifying agent in egg yolk [3]. The stability of pp 4M and pp 200 °C, together with the neutral charge of β-glucans, also suggested that the proteins contributing to their emulsification properties belong to the mannose-rich glycosylphosphatidylinositol (GPI)-anchored proteins family of yeast, proteins that yield emulsions resilient to pH variations (3–7) [22]. The improvement of the stability of pp 4M emulsion could be explained by the high energy associated to ultraturrax, which promotes the incorporation of short chain mannans and β-glucans into the emulsion. As with egg white proteins, high energy stirring probably caused the denaturation of pp 4M proteins, exposing hydrophobic groups and, ultimately, improving the emulsifying capacity. In the case of Sn 1M and Sn 4M, acidification decreased emulsions’ stability, a feature attributed to the different protein content, protein structure, and amino acid composition when compared with those found in pp 4M and pp 200 °C. *N*-linked mannoproteins are described to possess low solubility at acidic pH [10], resulting in the formation of aggregates [23], through the loss of the negative charges of glutamate (p*K_R_* = 4.25) and aspartate (p*K_R_* = 3.65). In this sense, mannoproteins become poorly dispersed along the aqueous phase and, due to the loss of electrostatic repulsion and oil droplets coalescence, result in lower emulsifying capacity [24]. However, *O*-linked mannoproteins exhibit high solubility at acidic pH due to their higher heterogeneity, leading to a low charge variation [10]. Nevertheless, xanthan gum was the emulsifier which yielded the emulsions of higher consistency, a feature related to the high thickening properties characteristic of this polysaccharide [25]. For all the other extracts, the similarity of the emulsions’ texture towards the reference emulsifiers makes them all highly valuable candidates for the replacement of egg yolks and modified starch in food emulsions.

The pudding-like structure of mayonnaise at a concentration of 3% of pp 4M might be attributed to the interaction of the extract with other ingredients, forming a semi-solid emulsion, as reported for the combination of mannoproteins with carboxymethylcellulose and xanthan gum [26]. The fact that this BSY extract and the remaining extracts also yielded a mayonnaise at 1% concentration supports the capability of BSY mannoproteins and β-glucans as emulsifiers in sauces [27,28], although resulting in products with greater fluidity than mayonnaises. These results suggest that the protein content and composition of the extracts are of high relevance in mayonnaise formulations.

## 4. Materials and Methods

### 4.1. Chemicals

All reagents and mayonnaise ingredients used in this work were of analytical grade and food grade, respectively. Sunflower oil (Fula, Portugal) and hen eggs were purchased at the local supermarket. Freeze-dried egg yolk was prepared after yolk separation. Alcohol vinegar, acetylated distarch adipate (pregelatinized modified starch, E1422), refined salt, fine granulated sugar, lemon juice concentrate, potassium sorbate (E202), lavender extract, and β-carotene (E160) were provided by Mendes Gonçalves (Golegã, Portugal).

### 4.2. Brewer’s Spent Yeast (BSY) Extracts

BSY (*Saccharomyces pastorianus*) was provided by Super Bock Group, SA (Leça do Balio, Portugal). BSY was subjected to a thermal autolysis at 60 °C for 2 h. The soluble fraction was removed by centrifugation (4696× *g*, 20 min, at 4 °C) and the obtained residue was then freeze dried, stored in a vacuum desiccator, and later used on both alkaline and subcritical water extractions.

#### 4.2.1. BSY Alkaline Extracts

Previously to the alkaline extraction, the freeze-dried BSY residue was suspended in 80% ethanol and boiled for 10 min. The alcohol insoluble residue (AIR) was then isolated by filtration using a porous plate funnel and washed three times with 95% ethanol and once with acetone. The AIR was then extracted with 1 M KOH or 4 M KOH, each in 20 mM NaBH_4_, using oxygen-free solutions to prevent peeling reactions. The alkali extractions were performed in a 1:60 (*w*/*v*) ratio at room temperature for 2 h, with continuous stirring under N_2_ atmosphere. The soluble material and residue were ultimately separated by centrifugation (24,700× *g*, 4 °C, 20 min), then neutralized with glacial acetic acid up to pH 5 and dialyzed in 12 kDa molecular weight cut-off membranes for salts and low molecular weight material removal. Given the formation of a precipitate during the neutralization of the soluble material, the polymeric material was centrifuged, yielding a precipitate (pp) and a supernatant (Sn) from the soluble material. During the dialysis of the final residue, a supernatant (Sn_FR) was collected and further concentrated under reduced pressure. All samples were frozen and freeze-dried, yielding the Sn 1M, pp 1M, Sn_FR 1M, Sn 4M, pp 4M, and Sn_FR 4M.

#### 4.2.2. BSY Subcritical Water Extracts

The freeze-dried BSY residue was resuspended in distilled water in a ratio of 1:6 (*w*/*v*) and placed in a Teflon reactor for 4 min at 180 °C and in another batch for 2 min, at 200 °C, under stirring in a microwave EthosSYNTH Labstation (maximum output, 1 kW, 2.45 GHz; Milestone Inc., Shelton, CT, USA). After extraction and depressurization, the suspensions were collected and centrifuged (24,700× *g*, at 4 °C, 20 min). The solubilized material was then recovered, and the polymeric material was isolated by ethanol precipitation at a concentration of 80% (*v*/*v*), yielding two subcritical water ethanol precipitates. Both precipitates and respective ethanol soluble material were then concentrated at reduced pressure to remove ethanol. These fractions were frozen and freeze-dried, yielding the Sn 180 °C, pp 180 °C, Sn 200 °C, and pp 200 °C.

### 4.3. Chemical Composition Analysis of BSY Extracts

Protein content was determined as total nitrogen content (N × 5.99) [29] by elemental analysis using a Truspec 630-200-200 equipment, combustion furnace temperature at 1075 °C, after burner temperature at 850 °C, and the detection mode used was thermal conductivity. The factor for estimation of protein content (N:P) was chosen based on consideration that the estimation of 2% chitin on BSY composition is not negligible.

Neutral sugars were released from the polysaccharides by acid hydrolysis and analyzed as their alditol acetates by gas chromatography [30] using a Perkin Elmer-Clarus 400 chromatograph with a split injector (split ratio 1:60) and a FID detector according to the method previously described [19].

Glycosidic linkage composition was determined by gas chromatography coupled to quadrupole mass spectrometry (GC-qMS) of the partially methylated alditol acetates (PMAA) [31] on a Shimadzu GCMS-QP2010 Ultra gas chromatograph, equipped with a HT-5 polycarborane-siloxane capillary column (Trajan Scientific, Victoria, Australia), according to the method described previously [19].

### 4.4. Emulsifying Properties of BSY Extracts

#### 4.4.1. Emulsifying Capacity and Stability

BSY extracts were added at a concentration of 22.5 mg/mL to 4 mL of water, then 6 mL of sunflower oil was added and hand-shaken or ultraturrax (13,500 rpm) stirred, for 30 s. The emulsifying capacity was then measured by the percentage of emulsion formed in the total volume (10 mL) at 24 h, the time needed for the complete separation of phases in the control (4 mL of water and 6 mL of oil). The stability of emulsions was evaluated by the emulsifying capacity after 1 month, comparing it with the emulsifying capacity at 24 h.

#### 4.4.2. Polysaccharide Composition of Emulsions

With the destabilization of emulsions, the separation of phases were observed, with the release of an aqueous phase from the emulsion, with higher density than the emulsion phase, and also the formation of a precipitate at the bottom was observed in some samples. Emulsion and aqueous phases from samples Sn 1M, Sn 4M, pp 4M, and pp 200 °C obtained using hand-shaking and ultraturrax were separated after 1 month storage. Aqueous phases were frozen and freeze-dried. Emulsion phases were submitted to a liquid–liquid extraction (LLE) with hexane (1:1), and the aqueous phases resulting from the LLE were frozen and freeze-dried. The freeze-dried material of each sample from the aqueous phases separated from the emulsion and the aqueous phases resulting from the LLE of emulsions were analyzed for their glycosidic linkage composition according to the method described in Section 4.3.

#### 4.4.3. Widefield Fluorescence Microscopy

Emulsions of xanthan gum and BSY extracts (pp 4M and pp 200 °C), performed using ultraturrax and after 1 month storage, were imaged (without any dilution) in a widefield fluorescence microscope (Zeiss Imager M2, Carl Zeiss, Germany), equipped with an EC Plan-Neofluar 5×/0.16 objective and an Axiocam 3MPix monochromatic camera. Micrographs were post-processed with ImageJ 2.1.0 (open-source image processing software, U. S. National Institutes of Health, Bethesda, MD, USA, 2020). Histograms have been obtained with the measurement of at least 500 droplets.

### 4.5. Texture Analysis of Mayonnaise Models with BSY Extracts Addition

Individually, xanthan gum, modified starch, freeze-dried yolk, and BSY extracts (Sn 1M, Sn 4M, pp 4M, and pp 200 °C) were solubilized in an aqueous solution composed of ingredients that provide acidity to mayonnaise, namely: potassium sorbate, lemon juice concentrate, and vinegar, in the same proportions of a standard recipe of mayonnaise, 1:2:36, respectively. The solution at pH 3 was stirred in an ultraturrax (IKA T25 basic Ultra Turrax) at the speed of 13,500 rpm for 30 s. Emulsifying capacity was measured as previously described in Section 4.4.1.

Texture analysis was carried out using a Texture Analyzer TA.XT Plus C (Stable Micro Systems, Cardiff, UK) with a 5 kg load cell. A back-extrusion test was performed using a compression disc of 35 mm diameter, which entered acrylic cylindrical containers (50 mm internal diameter and 75 mm height) containing 60 mL of sample at a depth of 35 mm. The test cycle was applied at a crosshead velocity of 0.5 mm/s until a depth of 70 mm, at which the probe returned. From the force–time curve, the empirical attributes as firmness, consistency, cohesiveness, and viscosity were obtained using Exponent Connect version 8,0,3,0 equipment software. Firmness was assessed as the maximum force of the compression cycle as the probe penetrated the sample [32]. The area of the curve established by the compression cycle, which finished at a depth of 70 mm, is taken as a measurement of consistency, meaning that the higher the area, the higher the consistency of the sample. When reaching a depth of 70 mm, the probe return produced a negative graph because of the resistance force provided by the sample. The maximum negative force reached was taken as an indicator of the cohesiveness of the samples, meaning that the more negative the value, the more cohesive the sample. The area of the curve obtained, defined as work of adhesion, defines the withdrawal resistance of the sample. In this sense, the greater the work of cohesion, the greater the consistency and viscosity of the sample.

### 4.6. Mayonnaises Formulation

Mayonnaises were produced by combining the ingredients in the following proportions (% *w*/*w*): 66% sunflower oil, 22% water, 5.6% egg yolk, 3.6% alcohol vinegar, 1.5% pregelatinized modified starch, 1.7% refined salt, 0.52% fine granulated sugar, 0.21% lemon juice concentrate, 0.13% potassium sorbate, 0.13% lavender extract, and 0.0041% β-carotene. The steps for ingredient mixing were: (1) water, β-carotene and all the powder ingredients using an IKA T25 basic Ultra Turrax at a speed of 9500 rpm; (2) addition of the egg yolk and mixing at the same speed; (3) slow addition of oil and mixing at a speed of 21,500 rpm; (4) addition of the vinegar and lemon juice concentrate at a speed of 13,500 rpm.

The BSY extracts were used as substitutes for egg yolks and modified starch in the mayonnaise formulation, accounting for 1 g of ingredient replacer, and compensation with ≈1 g of water was used as a reference for the moisture found in egg yolks.

The texture of standard mayonnaise and the mayonnaises with BSY extracts added were evaluated as described in Section 4.5.

## 5. Conclusions

Depending on the methodology used to extract BSY cell wall polysaccharides, extracts can be recovered with different emulsifying performances. In addition, it can be concluded that the extracts emulsifying properties are dependent from pH and the energy applied to form the emulsion. The relevant molecules for the BSY extracts emulsifying capacity at neutral pH are dependent on the energy applied. To obtain good emulsifying properties, less energy is necessary to apply in the presence of mannoproteins with high protein moiety, while high energy is necessary to emulsify oligosaccharide chains of (α1→2)-mannans and (β1→3)-glucans. The emulsion stability is related to the presence of small droplets, which is attained with the presence of mannoproteins exhibiting high protein moiety, which better prevent Ostwald ripening. The relevant molecules for the BSY extracts’ emulsifying capacity at acidic pH and high energy (ultraturrax stirring) are both mannoproteins with high protein moiety and short chains of (α1→2)-mannans, and also (β1→3)-glucans. Probably, mannoproteins contribute to decrease the surface tension at the interface with continuous phase and droplets, and (β1→3)-glucans could confer thickness to the continuous phase, conferring similar densities between bulk and droplets. The texture of the mayonnaise formulations was attained with 1/3 of the concentration of tensioactive molecules present in the standard formulation, when using mannoproteins with high protein moiety together with (β1→3)-glucans and (1→4)-glucans derived from glycogen. The substitution of egg yolk and modified starch (E1422) could be attained using BSY extracts, with those rich in mannoproteins obtained by alkali extraction, or those rich in mannooligosaccharides and (β1→3)-glucans obtained by subcritical water extraction.

## Figures and Tables

**Figure 1 molecules-28-03540-f001:**
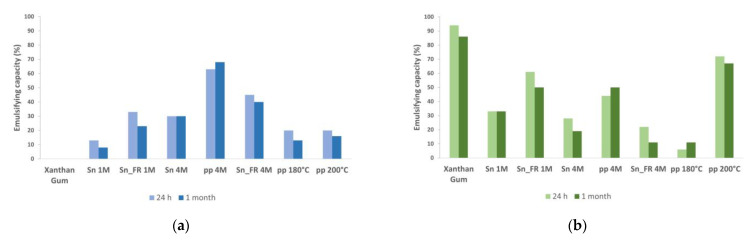
Emulsifying capacity at 24 h and at 1 month of xanthan gum and BSY extracts, at a concentration of 22.5 mg/mL, when applying: (**a**) hand-shaking; (**b**) ultraturrax.

**Figure 2 molecules-28-03540-f002:**
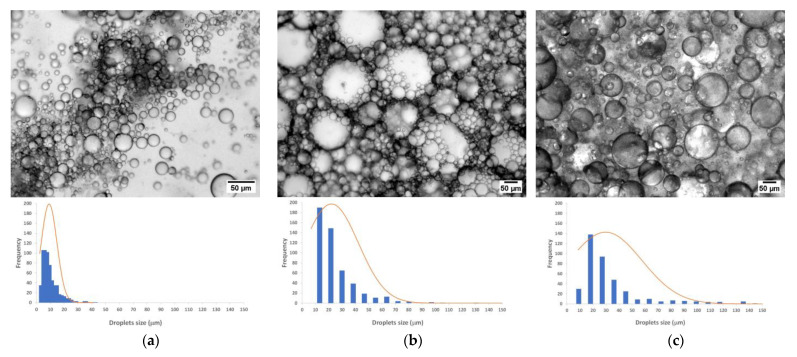
Optical contrast micrographs (5×), blue histogram and normal distribution (red curve) of droplets size (μm) by applying ultraturrax to the BSY extracts in water and vegetable oil (4:6), after 1 month: (**a**) pp 4M (50% emulsifying capacity); (**b**) pp 200 °C (67% emulsifying capacity); (**c**) xanthan gum (86% emulsifying capacity).

**Figure 3 molecules-28-03540-f003:**
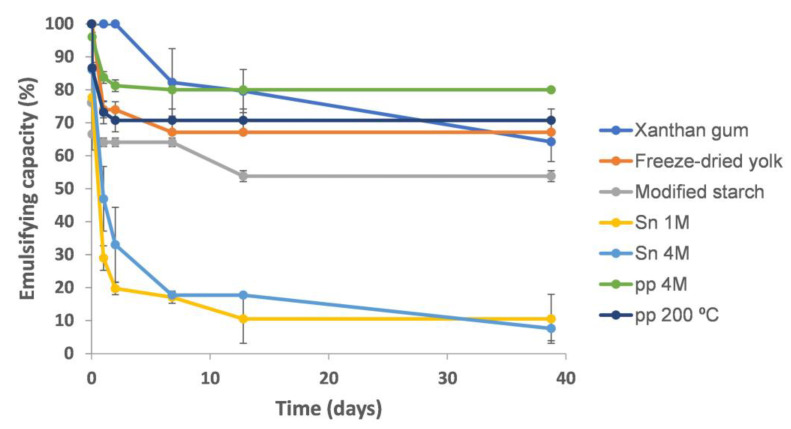
Emulsifying capacity of xanthan gum, freeze-dried yolk, modified starch, Sn 1M, Sn 4M, pp 4M and pp 200 °C, at a concentration of 2.5 mg/mL, in a mayonnaise emulsion model prepared with ultraturrax.

**Figure 4 molecules-28-03540-f004:**
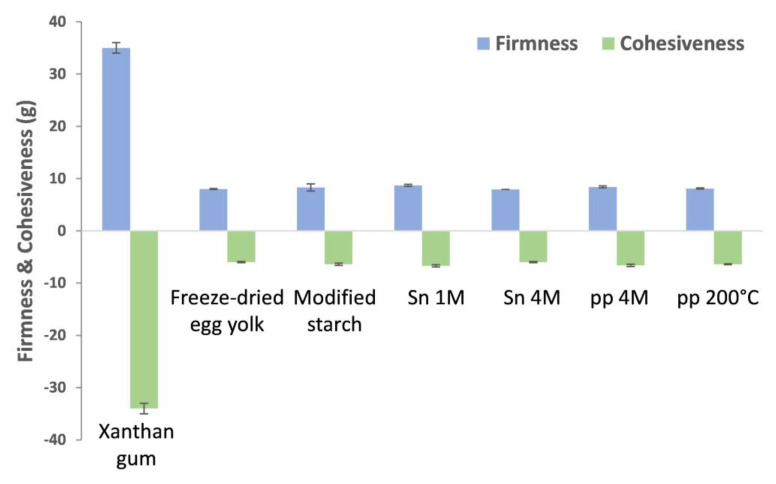
Firmness and cohesiveness of xanthan gum, freeze-dried yolk, modified starch, Sn 1M, Sn 4M, pp 4M, and pp 200 °C, at a concentration of 2.5 mg/mL, in a mayonnaise emulsion model prepared with ultraturrax (13,500 rpm).

**Figure 5 molecules-28-03540-f005:**
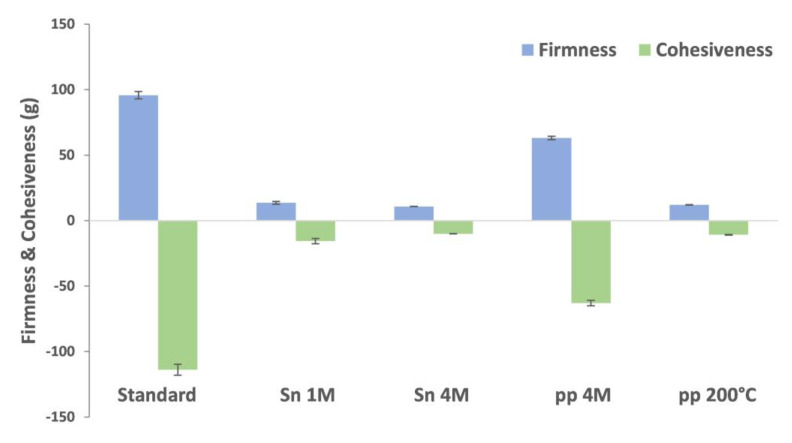
Firmness and cohesiveness of standard mayonnaise (3% of egg yolk + modified starch) and mayonnaises with addition of 1% BSY extracts (Sn 1M, Sn 4M, pp 4M, and pp 200 °C).

**Table 1 molecules-28-03540-t001:** Yields, carbohydrate, and protein composition of brewer’s spent yeast extracts.

Extraction		Samples	Yield (%)	Carbohydrates (mol%)		Total Carbohydrates (%)	Protein (%)
				Man	Glc		
		Autolysis Precipitate	100	30 ± 1	70 ± 1	30 ± 4	----
Alkaline	1M KOH	Sn	12	66 ± 2	34 ± 2	69 ± 2	23 ± 1
		pp	3	12 ± 2	88 ± 2	16 ± 2	27 ± 1
		Sn_FR	3	57 ± 0	43 ± 0	60 ± 2	14 ± 1
	4M KOH	Sn	17	77 ± 0	23 ± 0	45 ± 3	14 ± 2
		pp	13	12 ± 3	88 ± 3	21 ± 4	73 ± 0
		Sn_FR	4	10 ± 2	90 ± 2	79 ± 5	23 ± 1
SWE	180 °C	Sn	8	50 ± 3	50 ± 3	15 ± 1	32 ± 0
		pp	24	48 ± 1	52 ± 1	69 ± 7	32 ± 0
	200 °C	Sn	20	29 ± 1	71 ± 1	38 ± 0	32 ± 0
		pp	22	43 ± 1	57 ± 1	79 ± 9	32 ± 0

SWE—subcritical water extraction; Sn—supernatant; pp—precipitate; FR—final residue; Man—mannose; Glc—glucose.

**Table 2 molecules-28-03540-t002:** Glycosidic linkage composition (molar %) of brewer’s spent yeast extracts.

Glycosidic Linkage	Alkaline Extracts						SWE Extracts			
	Sn 1M	pp 1M	Sn_FR 1M	Sn 4M	pp 4M	Sn_FR 4M	Sn 180 °C	pp 180 °C	Sn 200 °C	pp 200 °C
t-Man	36.9	6.7	29.2	29.5	6.4	7.4	26.3	25.2	5.1	23.6
2-Man	16.1	3.9	8.9	17.3	3.4	3.6	25.8	12.5	3.8	9.8
3-Man	5.5	1.4	4.0	8.1	0.7	0.9	1.7	3.5	----	2.5
6-Man	0.4	0.4	0.6	1.6	0.2	0.2	1.2	1.6	----	0.8
2,6-Man	11.4	4.0	3.3	20.12	3.3	3.8	0.8	8.8	----	6.2
3,6-Man	----	----	0.2	2.0	----	----	0.3	----	-----	----
2,3,4,6-Man	0.3	0.4	----	----	0.2	----	----	----	3.4	0.2
**Total**	**70.6**	**17.4**	**46.2**	**78.5**	**14.2**	**15.8**	**56.1**	**51.5**	**12.3**	**43.1**
t-Glc	3.5	9.5	5.2	1.8	9.1	9.2	10.5	6.7	13.4	9.2
3-Glc	0.9	17.7	0.1	0.5	6.2	0.1	6.2	3.2	21.8	18.0
4-Glc	23.3	48.1	45.4	17.3	64.0	70.4	20.5	34.2	43.6	23.8
6-Glc	0.5	2.8	0.3	0.5	1.9	0.4	4.8	3.1	----	2.8
3,6-Glc	0.3	1.6	0.5	0.4	0.7	0.5	0.9	0.5	2.7	1.2
4,6-Glc	0.8	2.0	1.9	0.7	2.8	3.3	0.6	0.7	2.0	1.0
2,3,4,6-Glc	0.2	0.7	----	----	0.9	0.3	----	----	4.2	0.8
**Total**	**29.4**	**82.5**	**53.5**	**21.3**	**85.6**	**84.2**	**43.6**	**48.5**	**87.7**	**56.9**

SWE—subcritical water extraction; Sn—supernatant; pp—precipitate; FR—final residue; Man—mannose; Glc—glucose.

**Table 3 molecules-28-03540-t003:** Yields (%) of partition of the material after phase separation, emulsion, and aqueous phases in hand-shaken and ultraturrax emulsions, after 1-month storage.

Emulsion	Hand-Shaken			Ultraturrax		
	Emulsion Phase		Aqueous Phase	Emulsion Phase		Aqueous Phase
	LLE_Water	LLE_Hexane		LLE_Water	LLE_Hexane	
Sn 1M	21	0	79	21	19	60
Sn 4M	26	0	74	22	25	53
pp 4M	55	25	20	41	39	20
pp 200 °C	13	0	87	30	35	35

LLE_Water—water phase of emulsion obtained by liquid-liquid extraction (LLE); LLE_Hexane—Hexane phase of emulsion obtained by LLE.

**Table 4 molecules-28-03540-t004:** Glycosidic linkage composition (molar %) of brewer’s spent yeast extract’s emulsions (hand-shaken) from the material partitioned after phase separation, 1-month storage.

Glycosidic Linkage	Sn 1M		Sn 4M		pp 4M		pp 200 °C	
	Emulsion	Aqueous	Emulsion	Aqueous	Emulsion	Aqueous	Emulsion	Aqueous
t-Man	54	57	42	36	6	7	24	26
2-Man	16	19	7	12	2	2	9	11
3-Man	4	6	2	3	1	tr	2	2
6-Man	1	1	----	1	----	----	1	1
2,6-Man	3	6	----	4	tr	tr	5	5
2,3,4,6-Man	----	----	----	----	----	tr	----	----
**Total**	**78**	**88**	**51**	**56**	**9**	**9**	**41**	**45**
t-Glc	4	3	10	6	13	18	11	11
3-Glc	6	----	----	----	5	----	12	11
4-Glc	7	6	39	35	65	65	27	24
6-Glc	tr	tr	----	tr	1	1	4	3
3,6-Glc	tr	tr	----	1	2	1	2	2
4,6-Glc	1	1	----	2	4	4	2	2
2,3,4,6-Glc	3	1	----	tr	1	1	1	1
**Total**	**22**	**12**	**49**	**44**	**91**	**91**	**59**	**54**

Emulsion—emulsion water phase of liquid-liquid extraction; tr—traces.

**Table 5 molecules-28-03540-t005:** Glycosidic linkage composition (molar %) of brewer’s spent yeast extract’s emulsions (ultraturrax) from the material partitioned after phase separation, after 1-month storage.

Glycosidic Linkage	Sn 1M		Sn 4M		pp 4M		pp 200 °C	
	Emulsion	Aqueous	Emulsion	Aqueous	Emulsion	Aqueous	Emulsion	Aqueous
t-Man	63	54	55	46	18	16	51	42
2-Man	21	24	17	20	5	8	17	22
3-Man	7	6	4	5	1	2	3	5
6-Man	1	1	1	1	----	tr	1	3
2,6-Man	5	11	4	9	1	5	5	16
2,3,4,6-Man	tr	tr	tr	tr	----	tr	tr	tr
**Total**	**95**	**96**	**81**	**81**	**25**	**32**	**77**	**89**
t-Glc	1	tr	4	2	18	12	4	1
3-Glc	1	----	tr	----	16	----	13	7
4-Glc	2	2	12	13	35	42	1	1
6-Glc	----	tr	tr	1	----	1	1	1
3,6-Glc	tr	tr	1	1	2	2	tr	1
4,6-Glc	tr	tr	2	2	3	7	tr	tr
2,3,4,6-Glc	1	1	1	1	1	4	3	1
**Total**	**5**	**4**	**19**	**19**	**75**	**68**	**23**	**11**

Emulsion—emulsion water phase of liquid-liquid extraction; tr—traces.

## Data Availability

Not applicable.

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
