# Peer review of "Brewer’s Spent Yeast Cell Wall Polysaccharides as Vegan and Clean Label Additives for Mayonnaise Formulation"

_molecules, 2023, doi:10.3390/molecules28083540_

Round 1
Reviewer 1 Report
I am very grateful you for the invitation to review manuscript molecules-2311196 by Reis and coauthors "Brewer’s Spent Yeast Cell-Wall Polysaccharides as Vegan and Clean Label Additives for Mayonnaise Formulation”. This study verified the use of BSY mannoproteins as a clean label and vegan source of ingredients for the replacement of food additives and protein from animal sources. The work is interesting but needs several adjustments to increase the quality of the material.
Comments:
- Abstract, Lines 13-14: Please improve the problem presentation sentence. It's very simplistic.
- Abstract: The objective must be changed since the logical sequence is the extraction study and later the application in the product. Consider this for all text.
- Lines 19-23: Include verified numerical values for each analysis.
- Lines 30-31: Change the repeated keywords by different words from the title.
- Introduction: the formation of emulsions should be better detailed since the components will interfere with this formation chemistry.
- Lines 48-51: Include appropriate reference, considering market expectations for this.
- Line 65: Information regarding yeast production and generation of by-products must be entered (see the article https://doi.org/10.3389/frfst.2022.1074505 and others).
- Line 423: The authors should review this denomination since they used an ethanolic procedure prior to the alkaline treatment.
- Lines 430; 445; 470; 504: Standardize the number of rpm.
- 4.2.2 BSY subcritical water extracts: The same previous comment should be considered since the authors use a procedural protocol. This must be clear throughout the text.
- Except for the glucan extraction methods, the other items are well described and clear.
- Lines 78-80: The authors present it as a simple method of alkaline extraction, however, several steps are used and so they must be presented throughout the work, including the title. This comment should be considered for the entire text.
- Lines 123-129: The information is a repetition of what was presented in the material and methods item. Please show only results in this item.
- Lines 156-157: The information presented in the sentence should be transferred to the discussion item.
- Lines 295-296 and methodology: It is not clear whether animal products were completely replaced by extracts obtained from yeast. Please review.
- Lines 313-333: The information presented is the repetition of what has already been described in the results. Authors must clarify chemically what affects the process. In addition, a more in-depth discussion regarding the extraction methods applied should be included.
- Discussion: The entire item must be reviewed since little chemical and technological information is described for glucan extraction and mayonnaise production.
- Conclusion: A proper conclusion is not verified in the abstract and at the end of the discussion.
Author Response
Answers to the Round 1 Reviewer #1 comments
Manuscript ID: molecules-2311196
Brewer’s Spent Yeast Cell Wall Polysaccharides as Vegan and Clean Label Additives for Mayonnaise Formulation
Sofia F. Reis, Pedro A. R. Fernandes, Vítor J. Martins, Sara Gonçalves, Luís P. Ferreira, Vítor M. Gaspar, Diogo Figueira, Diogo Castelo-Branco, João F. Mano, Manuel A. Coimbra, and Elisabete Coelho
Comment 1:
” I am very grateful you for the invitation to review manuscript molecules-2311196 by Reis and coauthors "Brewer’s Spent Yeast Cell-Wall Polysaccharides as Vegan and Clean Label Additives for Mayonnaise Formulation”. This study verified the use of BSY mannoproteins as a clean label and vegan source of ingredients for the replacement of food additives and protein from animal sources. The work is interesting but needs several adjustments to increase the quality of the material.
- Abstract, Lines 13-14: Please improve the problem presentation sentence. It's very simplistic.”
Answer 1:
The authors thank the reviewer comment and suggestions to improve the manuscript. The problem presentation was improved.
Comment 2:
“- Abstract: The objective must be changed since the logical sequence is the extraction study and later the application in the product. Consider this for all text.”
Answer 2:
Text was amended to clearly state the use of extracted material for the application in ingredient replacement.
Comment 3:
“- Lines 19-23: Include verified numerical values for each analysis.”
Answer 3:
The numerical values for each analysis was included in the text.
Comment 4:
“- Lines 30-31: Change the repeated keywords by different words from the title.”
Answer 4:
We could not find any keywords that were repeated from the title.
Comment 5:
“- Introduction: the formation of emulsions should be better detailed since the components will interfere with this formation chemistry.”
Answer 5:
The introduction section was improved with the inclusion of a paragraph devoted to the emulsion constitution and formation.
Comment 6:
“- Lines 48-51: Include appropriate reference, considering market expectations for this.”
Answer 6:
A reference was introduced considering the growing demand for clean label products by consumers and market.
Comment 7:
“- Line 65: Information regarding yeast production and generation of by-products must be entered (see the article https://doi.org/10.3389/frfst.2022.1074505 and others).”
Answer 7:
Although not completely fitting the purpose of the sentence, the reference was included as suggested.
Comment 8:
“- Line 423: The authors should review this denomination since they used an ethanolic procedure prior to the alkaline treatment.”
Answer 8:
The denomination of section 4.2.1 as “BSY alkaline extracts” referrers to the obtention of extracts by alkaline extraction. In the present work, ethanol was used just to purify the polymeric material, as it solubilizes the lower molecular weight compounds that were discarded. Therefore, AIR preparation is only a sample preparation procedure to isolate the yeast cell walls. From them, the mannoproteins extraction was performed. The procedure was clarified with the information that the AIR had a previous treatment before alkaline extraction.
Comment 9:
“- Lines 430; 445; 470; 504: Standardize the number of rpm.”
Answer 9:
Centrifuge rpm values on lines 430 and 445 were standardized to x g values. The values on lines 470 and 504 are not changed as the ultraturrax apparatus rpm settings are unable to be standardized.
Comment 10: “- 4.2.2 BSY subcritical water extracts: The same previous comment should be considered since the authors use a procedural protocol. This must be clear throughout the text.”
Answer 10:
Rpm values were standardized to the respective x g values.
Comment 11:
“- Except for the glucan extraction methods, the other items are well described and clear.”
Answer 11:
This work uses different methodologies to extract yeast cell wall polysaccharides, not any specific polysaccharide such as glucans. For that reason, the description of the extractions performed are divided by the methodology used: alkaline and subcritical water, not by the polysaccharides, which were extracted by both methods. This idea was reinforced in the revised manuscript. Only after the assessment of the sugars and linkages composition of the extracts, could be termed as mannoprotein or glucan rich extracts.
The methodologies for extraction of glucans from yeasts should take into account the characteristics of the yeasts. For example, the glucans of BSY cell walls are poorly extracted because this BSY (S. pastorianus) is a very repitched yeast, leading to decrease (β1→3)-linked glucans whereas glycogen and (β1→4)-linked glucose residues, which conferred resistance to the yeast cell walls, increases (Bastos et al., 2015; Reis et al., 2023). Only when subcritical water is used, it is possible to extract some (β1→3)-linked glucans together with mannoproteins.
Comment 12:
“- Lines 78-80: The authors present it as a simple method of alkaline extraction, however, several steps are used and so they must be presented throughout the work, including the title. This comment should be considered for the entire text.”
Answer 12:
The text was clarified.
Comment 13:
“- Lines 123-129: The information is a repetition of what was presented in the material and methods item. Please show only results in this item.”
Answer 13:
The item was reformulated to give the context of the following results avoiding the repetition of material and methods information.
Comment 14:
“- Lines 156-157: The information presented in the sentence should be transferred to the discussion item.”
Answer 14:
The sentence was deleted. This information was already provided and discussed on the discussion section.
Comment 15:
“- Lines 295-296 and methodology: It is not clear whether animal products were completely replaced by extracts obtained from yeast. Please review.”
Answer 15:
Text was reviewed to clearly state the ingredient replacement by BSY extracts.
Comment 16:
“- Lines 313-333: The information presented is the repetition of what has already been described in the results. Authors must clarify chemically what affects the process. In addition, a more in-depth discussion regarding the extraction methods applied should be included.”
Answer 16:
The extractions were not optimized. Extraction methodologies were used as source of different extracts rich is components of yeast cell wall. Therefore, the discussion was driven to the composition of the extracts and not for the extraction procedures. Some discussion was included regarding the type of molecules that can be extracted with the procedures used.
Comment 17: “- Discussion: The entire item must be reviewed since little chemical and technological information is described for glucan extraction and mayonnaise production.”
Answer 17:
The glucan extraction, only attained with subcritical water, was included in the discussion. The mayonnaise production was already discussed concerning the textural analysis, the only analysis performed on mayonnaise. The discussion was centered on the relationship between chemical structure of molecules present in BSY extracts with emulsion formation and stabilization. The discussion led to the following achievements: the identification of the emulsifying molecules from BSY, which was based on model solutions, and their use in mayonnaise formulation as a proof of concept.
Comment 18:
“- Conclusion: A proper conclusion is not verified in the abstract and at the end of the discussion.”
Answer 18:
The conclusion at the end of the abstract and discussion was modified accordingly.

Reviewer 2 Report
The publication is interesting. A few minor remarks on Figure 1 there is no deviation of the measurement results the same in tables 3-5. If you can determine the measurement error then you should do it. I guess that the reagents used were of food grade purity this should be written directly.
Author Response
Answers to the Round 1 Reviewer #2 comments
Comment 1:
“The publication is interesting. A few minor remarks on Figure 1 there is no deviation of the measurement results the same in tables 3-5. If you can determine the measurement error then you should do it.“
Answer 1:
The authors thank the reviewer comment and suggestions. In figure 1, due to the low precision of the graduated falcon tubes (0.5 mL) the deviation was zero between the duplicates. Figures 3-5 have standard deviation, and it is shown as error bars in the figures.
Comment 2:
“I guess that the reagents used were of food grade purity this should be written directly.”
Answer 2:
Text was amended to clearly state the food grade origin of the mayonnaise formulation ingredients.
